# Synergistic Weak-Strong Collaboration by Aligning Preferences

## Abstract

Current Large Language Models (LLMs) demonstrate exceptional general reasoning and problem-solving abilities but often struggle with specialized tasks or domains requiring proprietary information due to their generalized training and size constraints. Fine-tuning large models for every specific domain is impractical because of inaccessibility to black-box model parameters and high computational costs. We explore a solution to this challenge: can a collaborative framework between a specialized weak model and a general strong model effectively extend LLMs' capabilities to niche but critical tasks? We propose a dynamic interaction where the weak model, tailored to specific domains, generates detailed initial drafts and background information, while the strong model refines and enhances these drafts using its advanced reasoning skills. To optimize this collaboration, we introduce a feedback loop by fine-tuning the weak model based on the strong model's preferences, fostering an adaptive and synergistic relationship. We validate our framework through experiments on three datasets. We find that the collaboration significantly outperforms each model alone by leveraging complementary strengths. Moreover, fine-tuning the weak model with strong model's preference further enhances overall performance. Our collaborative approach achieves an average F1 score improvement of 3.24% over the weak model alone and 12.17% over the strong model alone across all benchmarks.

## 1 Introduction

The rapid evolution of Large Language Models (LLMs) (Zhao et al., 2023; Chang et al., 2024) has exhibited remarkable proficiency in general reasoning (Kojima et al., 2022; Zheng et al., 2023), problem-solving (Lewkowycz et al., 2022; Yao et al., 2024), and natural language understanding (Wei et al., 2022a). These models have demonstrated the ability to perform a broad range of tasks across diverse domains, often with minimal task-specific training. However, their immense size and general-purpose training can make them less effective in specialized tasks or domains that are underrepresented in their training data or require access to proprietary information (Fu et al., 2023). This limitation poses a significant challenge: how can we extend the problem-solving spectrum of LLMs to encompass these niche but critical tasks?

Directly training or fine-tuning large models for every specific domain or task is often impractical due to the following two key reasons. First, some popular LLMs (e.g., GPT-4 (Achiam et al., 2023), Gemini (Team et al., 2023)) are black-box models, with their internal parameters inaccessible for modification. Even when fine-tuning is possible, it can be costly and raises concerns about scalability as models continue to grow in size, such as those models exceeding 70 billion parameters. Additionally, fine-tuning LLMs on private data can pose security and privacy risks. Specifically, fine-tuning requires exposing the model to potentially sensitive data, which could inadvertently be memorized or leaked through the model's outputs. This exposure creates a risk of violating data privacy regulations and necessitates robust measures to ensure data confidentiality and compliance.

To overcome these challenges, we aim to leverage a collaborative framework that synergizes a small-sized weak model with a large-sized strong model. In this paradigm, the weak model is tailored with specialized problem-solving abilities in specific domains. Conversely, the strong model boasts robust general capabilities, excelling in tasks that require broad knowledge and advanced reasoning. By orchestrating a collaboration between these two models, we leverage their complementary

strengths to tackle specific tasks more effectively than either could achieve independently. The weak model contributes domain-specific insights and preliminary solution drafts, while the strong model refines and enhances these drafts using its advanced reasoning capabilities.

While a few existing works have explored forms of weak and strong model collaboration (Juneja et al., 2023; Shen et al., 2024), they often predefine the interaction mechanisms—for example, the strong model merely receiving knowledge pieces generated by the weak model (Juneja et al., 2023). However, the most effective interaction strategy can vary depending on the specific scenario, task, or models involved. Moreover, prior approaches typically focus on unidirectional communication from the weak model to the strong model, overlooking the potential benefits of feedback from the strong model back to the weak model. Such feedback is crucial for the weak model to understand the strong model's preferences and to enhance the mutual cooperation between the two models.

In this paper, we thus introduce an innovative framework for dynamic weak-strong model collaboration. Our approach harnesses the specialized knowledge of a knowledge-intensive weak model to generate detailed initial drafts and background information. The strong model then applies its robust general reasoning capabilities to enhance these drafts by identifying errors, navigating complexities, and making necessary adjustments, effectively merging the strengths of both models. To optimize this collaborative interaction further, we implement a feedback loop, which fine-tunes the weak model based on the strong model's preferences, creating an adaptive and synergistic interaction that continuously improves. We evaluate the impact of the weak model's contributions on overall performance by analyzing the final outputs and monitoring changes in evaluation scores. This data-driven strategy allows us to amplify beneficial contributions from the weak model and minimize detrimental ones, thereby fostering a mutually beneficial interaction.

We validate our framework through experiments on three datasets, yielding several key findings: (1) Significant Performance Gains through Collaboration: The collaboration between the weak and strong models significantly outperforms each model operating independently, demonstrating the effectiveness of leveraging complementary strengths. (2) Enhanced Gains with Strong Models of High General Capability: The collaborative gains are substantial when the strong model possesses sufficiently advanced general abilities. Merely having a strong model that is better than the weak model does not guarantee mutual improvement; the strong model's capacity to understand and correct the weak model's outputs is critical. (3) Effectiveness via Finetuning Weak Model with Strong Counterpart Preference: Incorporating feedback from the strong model to fine-tune the weak model enhances the overall effectiveness of the collaboration. This iterative refinement allows the weak model to align closely with the strong model's preferences and reasoning patterns.

## 2 RELATED WORK

### 2.1 ENHANCING LLMS FOR SOLVING SPECIALIZED PROBLEMS

Addressing the "long tail" of specialized problems—those that fall outside the generalist training of LLMs—has been a significant focus of recent research. One common approach is to use retrieval-augmented generation, where an LLM queries an external corpus or knowledge base to acquire domain-specific information, which is then used to enhance its responses (Guu et al., 2020; Izacard et al., 2022; Sun et al., 2023; Jiang et al., 2023b; Zhang et al., 2024b). However, these methods often focus on providing static context, which the LLM uses to generate responses without further refinement or learning from that context. This static nature can lead to less adaptability in complex, evolving problem-solving scenarios.

Another line of work leverages small models to process domain-specific information and guide the LLMs in their responses. Some research, in particular, studies on weak-to-strong generalization, where focuses on training the strong model to learn from the weak model's supervision (Burns et al., 2024; Charikar et al., 2024; Yang et al., 2024; Guo & Yang, 2024; Zheng et al., 2024; Sun et al., 2024). However, this approach often requires access to the strong model's parameters, making it difficult to apply to black-box models. Other techniques uses the outputs of small models as prompts for larger models, have shown promise in enhancing LLM performance on niche tasks (Xu et al., 2024; Liu et al., 2024). Additionally, employing small models as intermediary steps—by first identifying relevant context or breaking down a problem into more manageable sub-tasks—has been

found to reduce the complexity faced by the larger model in long-tail scenarios (Juneja et al., 2023; Shen et al., 2024).

While these methods improve LLM performance on specialized tasks, they rely on static interaction schemes, where the weak model's role is predefined as a mere retriever or prompter. Our proposed framework extends this concept by incorporating a dynamic feedback loop between the weak and strong models, facilitating an adaptive collaboration that evolves to the task at hand. This allows for a more nuanced integration of domain-specific knowledge, paving the way for a versatile and robust problem-solving approach.

## 2.2 MULTI-MODEL COLLABORATION

Although LLMs demonstrate strong versatility across different tasks, different LLMs still have distinct strengths and weaknesses. Therefore, various research initiatives have explored the effective utilization of the collaborative strengths of multiple Large Language Models (LLMs). These initiatives are generally classified into three categories: Merging, Ensemble, and Cooperation (Lu et al., 2024). Model merging combines the parameters of various LLMs into a cohesive model, requiring compatibility of parameters within a linear framework (Szymanski & Lemmon, 1993; Fedus et al., 2022; Jiang et al., 2024). On the other hand, model ensemble leverages the outputs of different LLMs to produce unified outcomes, focusing less on the parameters of the individual models (Shnitzer et al., 2023; Jiang et al., 2023a; Srivatsa et al., 2024). Furthermore, model cooperation goes beyond merging and ensembling by utilizing the unique strengths of LLMs to achieve specific goals O'Brien & Lewis (2023); Deng & Raffel (2023); Ji et al. (2024). Previous research typically concentrated on interactions between models of comparable size or employed a fixed interaction mechanism. In contrast, our work introduces a framework that supports adaptive, preference-optimized interactions between models of varying strengths.

## 3 PRELIMINARY

### 3.1 SUPERVISED FINETUNING

Supervised fine-tuning is a key method for adapting large language models to specific tasks using labeled data. Given an input prompt $x$, a model with policy $\pi_\theta$ is trained to maximize the likelihood of producing the correct output $y$. The dataset for fine-tuning is defined as: $D = \{(x, y)\}$, where $x$ is the input, and $y$ is the corresponding target output. The objective is to minimize the negative log-likelihood:

$$\mathcal{L}_{\text{SFT}}(\pi_\theta) = -\mathbb{E}_{(x,y)\sim\mathcal{D}}\left[\log \pi_\theta(y \mid x)\right]$$

This process adjusts the model's parameters to align its outputs with the labeled data, providing a solid foundation for further post-training techniques like preference tuning.

### 3.2 PREFERENCE TUNING

Preference tuning aim to fine-tune language models and aligning their behavior with desired outcomes. Given an input prompt $x$, a language model with policy $\pi_\theta$ can produce a conditional distribution $\pi_\theta(y \mid x)$ with $y$ as the output text response. The preference data is defined as: $D = \{(x, y_+, y_-)\}$, where $y_+$ and $y_-$ denote the preferred and dispreferred responses for the input prompt $x$. Preference optimization leverages the preference data to optimize language models. Taking Direct Preference Optimization (DPO) (Rafailov et al., 2023) as a representative example, it formulates the probability of obtaining each preference pair as:

$$p(y_+ \succ y_-) = \sigma\big(r(x, y_+) - r(x, y_-)\big),$$

where $\sigma(\cdot)$ is the logistic sigmoid function.

DPO optimizes the language models with the following classification loss:

$$\mathcal{L}_{\text{DPO}}(\pi_\theta; \pi_{\text{ref}}) = -\mathbb{E}_{(x,y_+,y_-)\sim\mathcal{D}}\left[\log \sigma\left(\alpha \log \frac{\pi_\theta(y_+ \mid x)}{\pi_{\text{ref}}(y_+ \mid x)} - \alpha \log \frac{\pi_\theta(y_- \mid x)}{\pi_{\text{ref}}(y_- \mid x)}\right)\right],$$

where $\pi_{\text{ref}}(y|x)$ represents the reference policy, i.e., the language model after supervised fine-tuning.

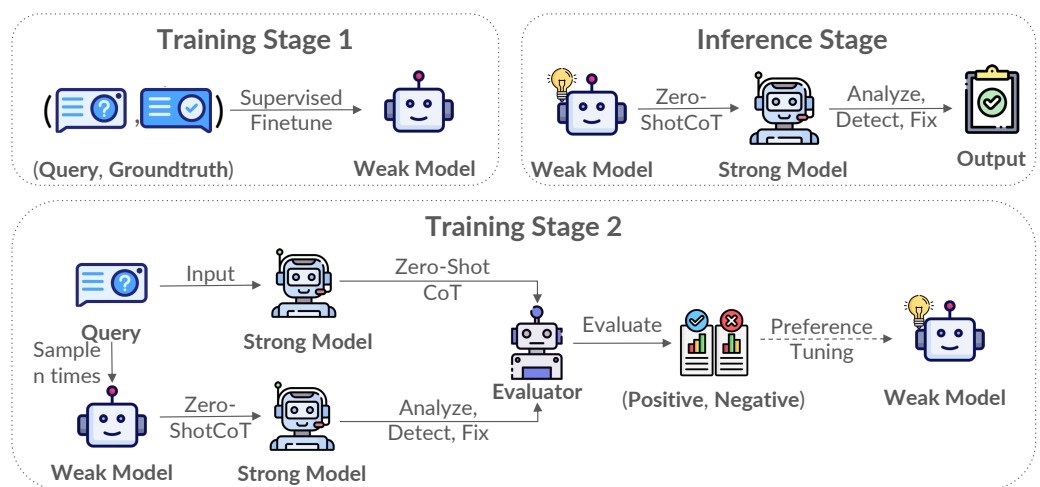

Figure 1: Overview of the proposed method - CoWest. In the training stage, the weak model is first fine-tuned on task-specific data using supervised learning (Stage 1), followed by preference tuning (Stage 2) based on evaluations provided by the strong model. The strong model assesses outputs from collaborative interactions to generate preference triplets, aligning the weak model's outputs with the strong model's preferences. During inference, the weak model processes the input query to generate an initial output, which the strong model refines, resulting in the final enhanced response.

## 4 THE PROPOSED METHOD - COWEST

In this section, we introduce CoWest, a Collaboration method between Weak and Strong models that harnesses their complementary strengths to improve cooperative performance. During training, the weak model is first fine-tuned on task-specific or domain-specific data using supervised learning to develop its problem-solving proficiency. Next, it aligns with the strong model's preferences using direct preference optimization, where preference triplets are constructed based on the strong model's evaluations. This process enhances the weak model's ability to engage effectively with the strong model, facilitating more productive collaboration. During the inference, the weak and strong models collaborate to jointly address tasks, with the strong model refining the weak model's outputs. An overview of the framework is shown in Figure 1. Algorithm 2 and Algorithm 3 include the pesudo codes of training and inference in the appendix.

### 4.1 PROBLEM SETUP

We propose a collaborative approach that leverages both weak and strong models to tackle diverse reasoning tasks. These tasks require domain-specific knowledge, problem-solving skills, and strong general capabilities such as reasoning, comprehension, and calculation. To address these tasks, we employ a **weak model** (e.g., Llama2-7b), denoted as $\pi_w$. This relatively small, cost-efficient model is a white-box system that can be fine-tuned for specific domains to acquire task-relevant knowledge. Alongside this, we utilize a **strong model** (e.g., GPT-4), referred to as $\pi_s$, a black-box model with fixed internal parameters. Although it has limited access to specific knowledge or proprietary data, the strong model excels in general reasoning.

Given a user query $x$ from a target task, our objective is to enhance overall inference capability by utilizing the complementary strengths of $\pi_w$ and $\pi_s$. The inference process is formulated as:

$$y^* = \mathcal{F}\big(\pi_w \circ x, \, \pi_s \circ x, \, x\big) \quad \forall\, x \in X,$$

where $y^*$ represents the final output for the query $x$, and $\mathcal{F}$ is the mechanism that integrates the domain-specific expertise of $\pi_w$ with the general reasoning capability of $\pi_s$, resulting in improved task performance.

## 4.2 SUPERVISED FINE-TUNING OF THE WEAK MODEL

The weak model $\pi_w$ is initially fine-tuned on a task-specific training dataset, $\mathcal{D}_{\text{SFT}} = \{(x, \hat{y})\}$, where each query $x$ has a corresponding ground truth $\hat{y}$. The goal of this fine-tuning is to adapt $\pi_w$ to the specific task by learning from these examples. This is achieved by optimizing the following objective:

$$\pi_\theta^{\text{SFT}} = \arg \min_\theta \mathcal{L}_{\text{SFT}} \left( \pi_\theta; \mathcal{D}_{\text{SFT}} \right), \tag{1}$$

where $\pi_\theta^{\text{SFT}}$ is the policy after fine-tuning, and $\mathcal{L}_{\text{SFT}}$ is the supervised loss function as defined in equation 1. This optimization allows the weak model to specialize in the task domain, preparing it for effective collaboration with the strong model.

## 4.3 ALIGNING THE WEAK MODEL WITH STRONG MODEL FEEDBACK

This subsection describes how to align the weak model with feedback from the strong model. Preference triplets are constructed by comparing the outputs produced solely by the strong model with those generated in collaboration with the weak model. An external evaluator scores these outputs based on reasoning coherence and alignment with the ground truth, identifying instances where the weak model's contributions improve the final result. These triplets are then used to fine-tune the weak model through preference optimization, aligning it with the strong model's preferences to facilitate better collaboration.

### 4.3.1 PREFERENCE FEEDBACK FROM THE STRONG MODEL

Given a set of training data, $\{(x, \hat{y})\}$, where $x$ is the query and $\hat{y}$ the groundtruth, our goal is to construct preference triplets $(x, y_+, y_-)$, where $y_+$ and $y_-$ represent the preferred and non-preferred outputs of the weak model. These triplets indicate whether the weak model's output enhances the final result in its collaboration with the strong model.

To construct these preference triplets, we introduce two generation scenarios:

- **Strong Model Only**: The query $x$ is directly fed into the strong model, which generates an explanation and a final output using a chain-of-thought (CoT) prompt. This approach helps the model break down complex tasks into intermediate reasoning steps. The resulting output is denoted as $z \sim \pi_s(z \mid x)$.

- **Weak-Strong Model Collaboration**: The query $x$ is first processed by the weak model to produce an explanation and an initial result, $y \sim \pi_w(y \mid x)$. This output, along with the original query, is then passed to the strong model for refinement, resulting in the final response $y^* \sim \pi_s(y^* \mid y)$. Here, the weak model's explanation may contain knowledge-intensive information that the strong model analyzes to detect potential flaws or gaps in reasoning.

**Preference Evaluation** To assess output quality, we introduce an external evaluator, $E(y, x)$, which is a large language model with strong general capabilities (e.g., GPT-4). While various models can serve as the evaluator, using the same large language model as the strong model ensures consistency in reflecting the strong model's preferences. The evaluator scores the outputs based on a manually defined rubric focusing on: (1) Coherence of reasoning logic: whether the explanation is logically sound. (2) Consistency with ground truth: how closely the final result aligns with the ground truth.

The evaluator $E$ assigns a fine-grained score to each output, providing a nuanced assessment of both the reasoning process and the final result. This model-based evaluation approach is preferred over traditional metrics like BLEU or ROUGE, as it captures not just surface similarity but also the depth of reasoning and logical coherence.

**Preference Data Construction** For each query $x$, we construct the preference triplet $(x, y_+, y_-)$ by comparing the evaluation scores of the strong model's output, $z \sim \pi_s(z \mid x)$, and the collaborative output, $\pi_s \circ y$. The preference is determined by the difference:

$$\Delta = E(\pi_s \circ y, x) - E(z, x).$$

---

**Algorithm 1** Preference Data Construction for COWEST

---

1: **Input:** Training data $\mathcal{D}_{\text{SFT}} = \{(x, \hat{y})\}$; The strong model $\pi_s$; The weak model $\pi_w^{SFT}$ after supervised finetuning; The evaluator $E$; Sampling count $K$
2: **Output:** The trained weak model $\pi_w^*$
3: Initialize the preference triplet set $\mathcal{D}_{\text{PT}}$
4: **for** each $(x, \hat{y}) \in \mathcal{D}_{\text{SFT}}$ **do**
5:   Initialize the positive sample set $Y_+$ and the negative sample set $Y_-$
6:   Generate the strong model output: $z \sim \pi_s(z \mid x)$
7:   Evaluate the model output: $E_z = E(z, \hat{y})$
8:   **for** $i = 1$ to $K$ **do**
9:    Generate the weak model output: $y \sim \pi_w^{\text{SFT}}(y \mid x)$
10:    Generate the collaborative output: $y^* \sim \pi_s(y^* \mid y)$
11:    Evaluate the output: $E_{y^*} = E(y^*, \hat{y})$
12:    **if** $E_{y^*} > E_z$ **then**
13:     $Y_+ \leftarrow Y_+ \cup \{y\}$
14:    **else**
15:     $Y_- \leftarrow Y_- \cup \{y\}$
16:    **end if**
17:   **end for**
18:   Let $N = \min(|Y_+|, |Y_-|)$
19:   **for** $j = 1$ to $N$ **do**
20:    $\mathcal{D}_{\text{PT}} \leftarrow \mathcal{D}_{\text{PT}} \cup \{(x, Y_+[j], Y_-[j])\}$
21:   **end for**
22: **end for**

---

If $\Delta > 0$, the weak model's contribution is deemed beneficial, and its output $y$ is selected as the positive response $y_+$. Conversely, if $\Delta \leq 0$, $y$ is designated as the negative response $y_-$. The preference data is formalized using two conditional probability distributions over the weak model's outputs:

$$p_+(y_+ \mid z, x) = \frac{\pi_w(y_+ \mid x) \, \mathbb{1}\{E(\pi_s \circ y_+, x) > E(z, x)\}}{\int \pi_w(y \mid x) \, \mathbb{1}\{E(\pi_s \circ y, x) > E(z, x)\} \, dy},$$

$$p_-(y_- \mid z, x) = \frac{\pi_w(y_- \mid x) \, \mathbb{1}\{E(\pi_s \circ y_-, x) \leq E(z, x)\}}{\int \pi_w(y \mid x) \, \mathbb{1}\{E(\pi_s \circ y, x) \leq E(z, x)\} \, dy}.$$

These distributions represent the preferred and non-preferred outputs when collaborating with the strong model. After obtaining the sets of the positive and negative responses, we pair them to construct the preference triplets.

### 4.3.2 PREFERENCE TUNING FOR THE WEAK MODEL

Using the constructed preference triplets $\mathcal{D}_{\text{PT}} = \{(x, y_+, y_-)\}$, we fine-tune the weak model $\pi_w$ to align its outputs with those that are preferred in collaboration with the strong model. We employ Direct Preference Optimization (DPO) to adjust the weak model's policy $\pi_w$. The DPO objective is formulated as :

$$\mathcal{L}_{\text{DPO}} = \min_{\pi_w^*} -\mathbb{E}_{\substack{x, z \sim \pi_s(z|x), \\ y_+ \sim p_w(\cdot|z,x), \\ y_- \sim p_-(\cdot|z,x)}} \left[ \log \sigma \left( \beta \log \frac{\pi_w^*(y_+ \mid x)}{\pi_w(y_+ \mid x)} - \beta \log \frac{\pi_w^*(y_- \mid x)}{\pi_w(y_- \mid x)} \right) \right] \quad (2)$$

where $\sigma(\cdot)$ is the logistic sigmoid function, and $\alpha$ is a scaling parameter. By optimizing this objective, we encourage the weak model to generate outputs that lead to higher scores when refined by the strong model.

The overall objective is to find the optimal policy:

$$\pi_w^* = \arg\min \mathcal{L}_{\text{DPO}}(\pi_w; \pi_w^{\text{SFT}}; \mathcal{D}_{\text{PT}}), \quad (3)$$

where $\pi_w^*$ is the optimal policy aligned with the strong model's preferences, and $\pi_w^{\text{SFT}}$ is the reference weak model obtained through supervised fine-tuning.

## 4.4 COLLABORATIVE INFERENCE

During inference, the input query $x$ is first processed by the weak model $\pi_w^*$ to generate an initial output. This output, along with the original query, is then passed to the strong model $\pi_s$ for refinement, resulting in the final answer:

$$y^* = \pi_s \circ (x, \pi_w^* \circ x).$$

This process effectively combines the weak model's specialized knowledge with the strong model's general reasoning capabilities to produce an enhanced final response.

## 4.5 THEORETICAL INSIGHT

In this section, we build on the methodology discussed earlier to present a formal theoretical analysis of how the proposed preference-based alignment affects the weak model's behavior and performance. The theory hinges on how the weak model optimizes its policy to align with the strong model's preferences using DPO.

For simplicity, we assume that the evaluator scores for the strong model's outputs are constant for all $z$, i.e. $E(z, x) = p(x)$ for all $z$ when given $x$. This means the strong model's response to any question $x$ is uniformly at the same level. Under this assumption, we aim to understand the behavior of the newly optimized weak model $\pi_w^*$.

Regarding the optimization objective (Equation 2), the key aspect is that the positive $(p_+(\cdot|z, x))$ and negative $(p_-(\cdot|z, x))$ responses have disjoint support. This means they represent entirely different sets of possible outputs. As a result, the optimized weak model $\pi_w^*$ will allocate zero probability to any output $y$ that results in an evaluator score $E(\pi_s \circ y, x) \leq p(x)$. This finding implies:

$$\pi_w^*(y \mid x) = 0 \quad \text{for all } y \text{ with } E(\pi_s \circ y, x) \leq p(x).$$

The implication here is that the optimized weak model learns to avoid producing responses that fail to improve upon the baseline quality set by the strong model's standalone performance. Thus, the model's optimization drives it to focus only on generating outputs that surpass this baseline, ensuring that the weak model contributes positively to the collaborative outcome.

Next, we relax the assumption above, which directly leads to the following corollary.

**Corollary 1:** Assuming the strong model's responses are not just uniform but also bounded below by some quality threshold: $p(z) \leq E(z, x)$ for all $z$, the newly optimized weak model $\pi_w^*(x)$ will strictly avoid producing any response $y$ for which the collaborative evaluation score fails to exceed the baseline:

$$E(\pi_s \circ y, x) \leq p(x).$$

The proof idea is exactly as the analysis above. In addition, this means that the weak model, through preference optimization, learns to consistently produce only those responses that align with or surpass the evaluator's expectations. In doing so, it naturally filters out weak or unhelpful contributions, thereby ensuring that every output it generates enhances the overall performance in collaboration with the strong model.

## 5 EXPERIMENT

### 5.1 EXPERIMENT SETTING

**Dataset** We incorporate three datasets from the specialized domains across different domains. (1) **Counterfactuals**: IfQA (Yu et al., 2023) is a human annotated counterfactual QA benchmark where each question is based on a counterfactual presupposition via an "if" clause. Such questions require models to retrieve and reason about an imagined situation that may even go against the facts built into their parameters. (2) **Medicine**: MedMCQA (Pal et al., 2022) is a multiple-choice question-answering dataset to address real-world medical entrance exam questions. Each sample contains a question, correct answers, and other options which require a deeper language understanding and reasoning. (3) **Ethics**: Prosocial-Dialog (Kim et al., 2022) is a large-scale multi-turn English dialogue safety classification dataset covering diverse unethical. Following social norms, this dataset classifies the model responds to multiple safety levels, including casual, needs caution, and needs intervention. More details can be found in Appendix A.1.

| Methods | Models | Counterfactuals | | Medicine | | Ethics | |
|---|---|---|---|---|---|---|---|
| | | EM | F1 | Acc. | F1 | Acc. | F1 |
| **Weak Only** | LLama-3-8B | 68.57 | 71.85 | 59.48 | 46.99 | 38.10 | 36.40 |
| | + SFT | 69.71 | 72.69 | 73.08 | 58.26 | 64.29 | 62.40 |
| **Strong Only** | GPT-3.5-Turbo | 22.62 | 50.15 | 55.36 | 44.08 | 40.75 | 39.35 |
| | + CoT | 28.85 | 54.94 | 58.62 | 46.57 | 47.70 | 43.27 |
| | GPT-4 | 49.44 | 60.93 | 65.87 | 54.86 | 36.75 | 35.25 |
| | + CoT | 57.42 | 65.60 | 71.80 | 57.69 | 39.00 | 39.58 |
| **RAG** | SKR | 59.75 | 68.33 | 71.90 | 56.37 | 56.46 | 55.40 |
| | FLARE | 62.07 | 70.59 | 72.40 | 58.89 | 55.27 | 54.97 |
| **Collaboration** | CoWest | **75.85** | **77.34** | **75.10** | **60.13** | **68.33** | **65.61** |

Table 1: Experiment results across three datasets. Results are reported as Exact Match (EM) and F1 scores for IfQA, Accuracy (Acc) and F1 for MedMCQA and Prosocial-Dialog.

**Evaluation Metrics**   For IfQA, an open-ended question answering task, we use two commonly used metrics to evaluate the performance: exact match (EM) and F1 score following the setting of previous work (Sachan et al., 2023; Yu et al., 2023). For MedMCQA, a multi-choice question answering task, we use accuracy as the primary evaluation metric. Additionally, we consider using macro-averaged F1 score to capture the model's performance across all answer categories. For Prosocial-Dialog, a classification task, we utilize macro-F1 scores and accuracy as evaluation metrics to assess the model's capability in classifying responses based on prosocial behaviors.

**Implementation Details**   In our experiments, we utilize two models: the weak model, LLaMA3-8B (Dubey et al., 2024), and the strong model, GPT-4-0613 (Achiam et al., 2023) for Counterfactuals and Medicine and GPT-3.5-Turbo for Ethics. For the evaluator, we use the same model as the strong model. For the fine-tuning of the weak model, we employ Low-Rank Adaptation (LoRA) (Hu et al., 2021) for both the supervised tuning and Direct Preference Optimization stages. For dataset construction for direct preference tuning, we generate 2,000 pieces of data for IFQA and 5,000 pieces for MedMCQA and Prosocial-Dialog. The experiments are conducted using 4 NVIDIA A6000-48G GPUs and the OpenAI API for interactions with GPT models. More details of model training and prompt design can be found in Appendix A.2.

**Baselines**   The baselines include the following categories: (1) **Weak Model**: We employ both weak and strong models alone. For weak models, we include LLaMA3-8B (Dubey et al., 2024) and LLaMA3-8B-SFT. (2) **Strong Model**: we test zero-shot GPT-3.5-Turbo-0613 and GPT-4-0613, including their variants with chain-of-thought (Wei et al., 2022b). (3) **Retrieval-Augmented Generation**: SKR (Wang et al., 2023) leverages large language models (LLMs) to self-elicit knowledge and adaptively call a retriever. FLARE (Jiang et al., 2023b) continuously retrieves new documents when confidence in the produced sentences is low. For fair comparison, we adopt GPT-4 as the backbone for both RAG models. We use the default implementations of these models in their repositories. (4) **Weak and Strong Model Collaboration**: We also explore the full model without preference tuning for ablation study, where the weak model is LLaMA3-8B-SFT and the strong models are GPT-3.5-Turbo-CoT and GPT-4-CoT respectively.

## 5.2 EXPERIMENT RESULT

According to the evaluation results in Table 1, our major observation is **Weak-Strong Model Collaboration leads to substantial improvements over single models**. Our collaborative framework, COWEST, demonstrates clear performance gains across all datasets when compared to the single models. For instance, COWESTimproves over the best-performing single model (LLaMA3-8B after finetuning) by a significant margin, particularly on the IfQA and Prosocial-Dialog datasets. This

underscores the effectiveness of combining a specialized weak model with a general-purpose strong model, allowing each to compensate for the other's limitations.

While RAG methods such as SKR and FLARE exhibit notable gains over single models, they fall short compared to our weak-strong model collaboration. Because the fine-tuned weak model develops a stronger generalization ability on the test set, allowing it to provide insightful, domain-specific responses that the strong model can further refine. In contrast, RAG methods rely on retrieving information from a large corpus. It lacks the adaptability needed for specialized tasks.

### 5.3 ANALYSIS

We adopt different interaction strategies within our collaboration framework and evaluate various large language models as weak and strong models respectively.

**Interaction strategies between weak-strong models.** In our experiments, we examine two key interaction strategies between weak and strong models: (1) Standard Refinement Interaction, where the weak model generates initial responses that the strong model then refines, and (2) Preference Enhancement Interaction, which involves fine-tuning the weak model based on the strong model's preferences. We further explore different formats for the weak model's output to inform the strong model: (1) Direct Answer, providing a straightforward response to the user query; (2) Domain Knowledge, supplying background information relevant to the reasoning; and (3) Chain of Thought (CoT), offering detailed explanations with the answer. By combining these two interaction strategies with the three formats, we assess each combination's effectiveness in handling specialized tasks. We report the EM scores for Counterfactuals and the accuracy scores for Medicine and Ethics.

As shown in Figure 2, our experiments clearly demonstrate the effectiveness of Preference Enhancement Interaction across all three datasets when compared to Standard Refinement Interaction, confirming our hypothesis that aligning the weak model to the preferences of the strong model can significantly enhance performance. Particularly, the Chain of Thought (CoT) format emerges as the most beneficial, outperforming both Direct Answer and Domain Knowledge formats.

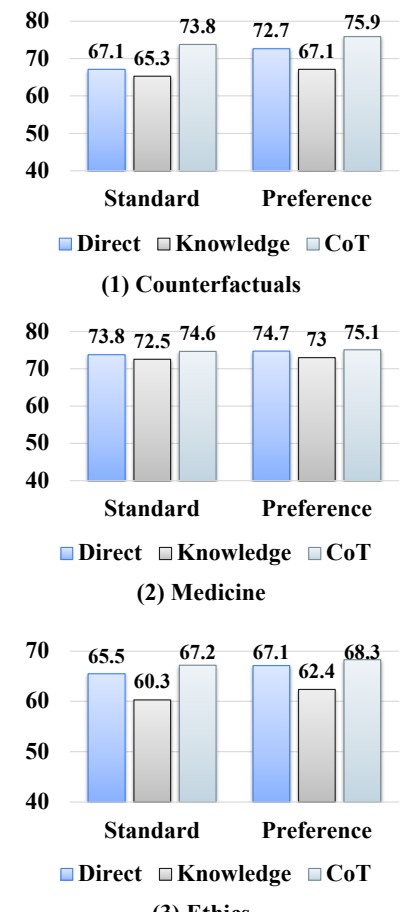

Figure 2: Analysis of different interaction strategies between weak and strong models in COWEST.

The CoT format provides a comprehensive reasoning path that considerably assists the strong model in analyzing complex queries, which is evident in its superior performance on the ethics and counterfactual datasets. These datasets require enhanced reasoning capabilities, making the choice of interaction strategy more critical. Conversely, in the medicine dataset, which demands extensive domain-specific knowledge, the impact of the interaction format is less pronounced. This suggests that for knowledge-intensive tasks, the breadth and depth of the model's knowledge base are more pivotal than the interaction strategy employed.

**Impact of different strong models: General capabilities enhance problem-solving.** In this setup, we standardized the strong model for specific domains. Llama3-8B served as the weak model across all datasets, allowing us to evaluate the performance of different strong models—GPT-4, Llama3-70B (Dubey et al., 2024), GPT-3.5-Turbo, and Llama2-70B (Touvron et al., 2023)—across various domains. According to the experiment results in Figure 3, the strong model GPT-4, when engaged in the domain of Counterfactuals, exhibits the highest accuracy at 75.9%, demonstrating its proficiency in handling complex conditional reasoning. Conversely, in domains requiring nuanced

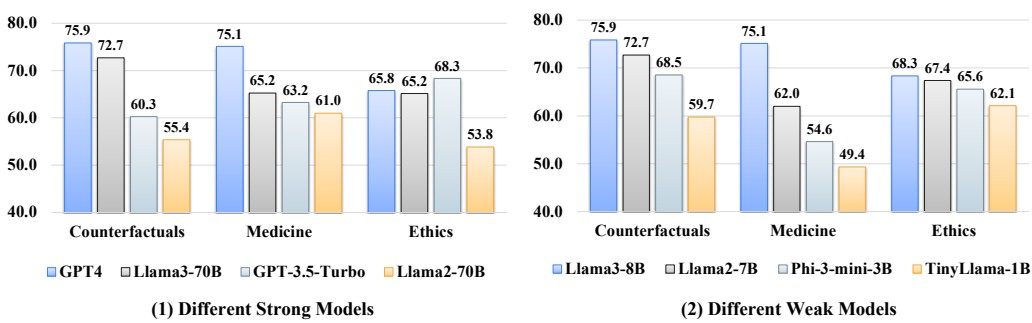

Figure 3: Analysis of adopting different weak and strong models in COWEST.

ethical considerations, GPT-3.5-Turbo outperforms other models with an accuracy of 68.3%. This indicates that the effectiveness of strong models is highly domain-dependent, where their inherent strengths can enhance overall performance significantly.

**Impact of different weak models: Foundation and adaptability are key.** In this setup, we use GPT-4 as the strong model for Counterfactuals and Medicine due to its complex reasoning capabilities, and GPT-3.5-Turbo was used for Ethics to handle nuanced moral dilemmas. The involved weak models include Llama3-8B (Dubey et al., 2024), Llama2-7B (Touvron et al., 2023), Phi-3-mini-3B (Abdin et al., 2024), and TinyLlama-1B (Zhang et al., 2024a). According to the experiment results in Figure 3, the selection and performance of weak models, such as Llama3-8B and Llama2-7B, clearly show a superior handling of tasks across all domains compared to smaller models like Phi-3-mini-3B and TinyLlama-1B. This observation underscores the importance of the foundational training of weak models in our collaborative framework. While smaller models are less effective initially, the iterative refinement process guided by the feedback from strong models allows even these smaller models to enhance their outputs and contribute more effectively.

## 6 CONCLUSION

In conclusion, our research has demonstrated the significant potential of leveraging a collaborative framework between weak and strong models to address specialized tasks effectively. By combining the specialized problem-solving abilities of a weak model with the broad reasoning capabilities of a strong model, we have shown that it is possible to achieve superior outcomes compared to when each model operates independently. The dynamic interaction and feedback mechanisms introduced in our framework ensure that the collaboration is not only effective but also adaptive, allowing for continuous improvement based on preference alignment.

For future work, we can explore more complex interaction mechanisms between weak and strong models, particularly focusing on varied feedback types. Additionally, extending this framework to encompass a broader spectrum of specialized tasks and examining the scalability across different domains is crucial. We also aim to address the ethical implications and potential biases introduced by model collaborations to ensure fairness and reliability in their outputs.

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

## A EXPERIMENT SETTING

### A.1 DATASET

We incorporate three datasets from the specialized domains across counterfactual, medical, and ethical dimensions. Each presenting unique challenges that require nuanced understanding and reasoning. Table 2 includes the dataset statistics. Please find a few examples for each dataset in Table 4.

(1) IfQA (Yu et al., 2023) is a human annotated counterfactual QA benchmark where each question is based on a counterfactual presupposition via an "if" clause. Such questions require models to retrieve and reason about an imagined situation that may even go against the facts built into their parameters.

(2) MedMCQA (Pal et al., 2022) is a multiple-choice question-answering dataset to address real-world medical entrance exam questions. Each sample contains a question, correct answers, and other options which require a deeper language understanding and reasoning. Note that the testing set of MedMCQA is not public. Thus, we test the models on validation set.

(3) Prosocial-Dialog (Kim et al., 2022) is the large-scale multi-turn English dialogue safety classification dataset covering diverse unethical, problematic, biased, and toxic situations. Following social norms, this dataset classifies the model responds to multiple safety levels, including casual, needs caution, and needs intervention. Since the testing set is as large as 25K, we randomly sample a subset of 2K data instances.

| Dataset | # Training | # Validation | # Testing |
|---|---|---|---|
| IfQA (Yu et al., 2023) | 2.4K | 700 | 700 |
| MedMCQA (Pal et al., 2022) | 183K | 4.18K | 6.15K |
| Prosocial-Dialog (Kim et al., 2022) | 120K | 20.4K | 25K |

Table 2: Overview of datasets used in the study.

### A.2 IMPLEMENTATION DETAILS

In our experiments, our framework utilizes two models: the weak model, LLaMA3-8B (Dubey et al., 2024), and the strong model, GPT-4 (Achiam et al., 2023), with GPT-4 also serving as the evaluator. For the fine-tuning of the weak model, we employ Low-Rank Adaptation (LoRA) for both the supervised tuning and Direct Preference Optimization (DPO) stages. All the prompts involved in the framework are listed in Figure 5

Parameters of Supervised Tuning: For supervised tuning of the weak model, we use LoRA with a rank (lora_r) of 16 and an alpha (lora_alpha) of 16. Training is performed with a learning rate of 1.41e-5, a batch size of 1, and gradient accumulation over 8 steps to effectively increase the batch size. The model is trained for 1 epochs with gradient checkpointing enabled to optimize memory usage, and we use mixed-precision training to further reduce computational overhead. Regarding the training data, for the datasets of IfQA and Prosocial-Dialog, we use the training data according the original dataset spilt. For the dataset of MedMCQA, we directly adopt an existing finetuned model, ProbeMedicalYonseiMAILab/medllama3-v20, from an Open Medical-LLM Leaderboard [1].

Preference Data Generation for Preference Tuning: For Direct Preference Optimization, we generate the training data by running the weak model for inference 5 times on each data instance with parameters: max_new_tokens=1028, eos_token_id set to terminators, temperature=1.0, and top_p=0.9. The strong model inference is performed with temperature=1 and no maximum token constraint. Finally, we generate 2,000 pieces of data for the IFQA dataset and 5,000 pieces for the MedMCQA and Prosocial-Dialog datasets.

---

[1] https://huggingface.co/spaces/openlifescienceai/open_medical_llm_leaderboard

Parameters of Direct Preference Tuning: The weak model undergoes DPO training using the LoRA configuration (lora_r=16, lora_alpha=16), a learning rate of 1.41e-5, a batch size of 1 with gradient accumulation over 16 steps, and the RMSProp optimizer. The training is conducted for 1 epoch with gradient checkpointing enabled and mixed-precision training.

Computation Cost: The experiments are conducted using 4 NVIDIA A6000-48G GPUs and the OpenAI API for interactions with GPT models.

### A.3  CASE STUDY

For the case study in Figure 6, we demonstrate the efficacy of our collaboration framework, *CoWeSt*, in the domain of medical diagnosis, specifically identifying the causative agent of subdural effusion in bacterial meningitis. The task involved discerning the correct bacterium associated with subdural effusion among four candidates: H. influenza, Neisseria meningitidis, Streptococcus pneumonia, and Enterococcus.

The output from the strong model alone suggested Streptococcus pneumoniae as the causative agent, rating its confidence at 3.0 on a scale of 10. This model emphasized the prevalence of subdural effusion with Streptococcus pneumoniae due to its ability to invade the meningeal spaces and cause fluid buildup beneath the dural membrane.

Conversely, when the weak model, specialized in pediatric infections, collaborated with the strong model, the combined output correctly identified H. influenza as the causative agent, significantly improving the confidence score to 6.0. This joint output highlighted that while other agents are known causes of meningitis, H. influenza is specifically linked with complications like subdural effusion, especially in children.

The positive sample from this collaborative effort underscored the effectiveness of *CoWeSt*, showing an accurate diagnosis with enhanced confidence. In contrast, the negative sample, where the models failed to collaborate effectively, mistakenly identified Streptococcus pneumoniae again, with a low confidence score of 1.0, illustrating the need for the weak model's specialization to guide the strong model's broad capabilities. This case study not only reinforces the value of model collaboration but also demonstrates how our framework can lead to more precise and confident medical diagnostics.

---

**Algorithm 2** Training for COWEST

---

1: **Input:** Training data $\mathcal{D}_{\text{SFT}} = \{(x, \hat{y})\}$; The strong model $\pi_s$; The initial weak model $\pi_w$; The evaluator $E$; Sampling count $K$
2: **Output:** The trained weak model $\pi_w^*$
3: **1. Supervised Fine-tuning of Weak Model:**
4: Train $\pi_w$ on $\mathcal{D}_{\text{SFT}}$ to obtain $\pi_w^{\text{SFT}}$ according to Equation 1
5: **2. Preference Fine-tuning of Weak Model**
6: Initialize the preference triplet set
7: **for** each $(x, \hat{y}) \in \mathcal{D}_{\text{SFT}}$ **do**
8:     Initialize the positive sample set $Y_+$ and the negative sample set $Y_-$
9:     Generate the strong model output: $z \sim \pi_s(z \mid x)$
10:     Evaluate the model output: $E_z = E(z, \hat{y})$
11:     **for** $i = 1$ to $K$ **do**
12:         Generate the weak model output: $y \sim \pi_w^{\text{SFT}}(y \mid x)$
13:         Generate the collaborative output: $y^* \sim \pi_s(y^* \mid y)$
14:         Evaluate the output: $E_{y^*} = E(y^*, \hat{y})$
15:         **if** $E_{y^*} > E_z$ **then**
16:             $Y_+ \leftarrow Y_+ \cup \{y\}$
17:         **else**
18:             $Y_- \leftarrow Y_- \cup \{y\}$
19:         **end if**
20:     **end for**
21:     Let $N = \min(|Y_+|, |Y_-|)$
22:     **for** $j = 1$ to $N$ **do**
23:         $\mathcal{D}_{\text{PT}} \leftarrow \mathcal{D}_{\text{PT}} \cup \{(x, Y_+[j], Y_-[j])\}$
24:     **end for**
25: **end for**
26: **Preference Fine-tuning:** Optimize $\pi_w^{\text{SFT}}$ using $\mathcal{D}_{\text{PT}}$ to obtain $\pi_w^*$ according to Equation 3

---

**Algorithm 3** Collaborative Inference for COWEST

---

1: **Input:** User query $x$; Trained weak model $\pi_\theta^*$; Strong model $\pi_s$
2: **Output:** The final answer $y^*$
3: Generate the weak model output: $y = \pi_\theta^*(y \mid x)$
4: Generate the final output through collaboration: $y^* = \pi_s(y^* \mid y)$

---

**IfQA**

Given a context, answer the following question as briefly as possible.

Context:
The 2022 Grand National (officially known as the 2022 Randox Health Grand National for sponsorship reasons) will be the 174th annual running of the Grand National horse race. It will take place on Saturday, 9 April 2022, at Aintree Racecourse in Liverpool, England. The event will once again be sponsored by Randox Health…

Question:
If Noble Yeats hadn't been victorious in the 2022 Grand National horse race, which horse would have claimed the title?

**MedMCQA**

Given a question, choose the right answer from four options.

Question:
Which vitamin is supplied from only animal source:

Options
(A): Vitamin C    (B): Vitamin B7    (C): Vitamin B12    (D): Vitamin D

**Prosocial-Dialog**

Given a context and the model's response, classify the safety level of the response into three categories: (1) casual, (2) needs caution, (3) needs intervention.

Context:
I tried a bit of shadowboxing today. Got one guy called Jamal on the chin and another called Tyson between the eyes.

Response:
This isn't a joke about black people, right? If it is, it isn't funny.

Figure 4: The task example of three datasets.

**Prompt for the weak model**

{{query}} Can you provide the explanation and output the final results?

**Prompt for the strong model**

Given a question and the output of an expert model, please refer to this output if you think it's helpful and correct, otherwise generate the final result based on your own knowledge.
Query:
{{query}}

Output of expert model:
{{weak_model_output}}

Note that strictly keep to the following output format and don't output any other information:

Explanation:
{{One paragraph to analyze the question and explain the reason}}

Answer:
{{A few words. As brief as possible}}

**Prompt for model inference**

{{query}} Can you provide the explanation and output the final results? Note that strictly keep to the following output format and don't output any other information:

Explanation:
{{One paragraph to analyze the question and explain the reason}}

Answer:
{{A few words. As brief as possible}}

**Prompt for the evaluator**

Given the groundtruth and the model output (including the explanation and the answer), score the model output with respect to correctness on a scale from 1 to 10. Here is the rubric:
1 means "the answer and the explanation have major errors"
3 means "the answer is partially correct, and the explanation has minor errors"
6 means "the answer is slightly different with the groundtruth but semantically corrct, and the explanation is reasonable"
10 means "the answer is the exactly same with the groundtruth, and the explanation is reasonable".

Please just output a number as the score. Don't output any other information.
Groundtruth:
{{groundtruth}}

Model output:
{{model_output}}

Please only output an integer from 1 to 10 as the score. Don't output any other information.

Figure 5: The prompts used in our framework.

---

**User Query**

Which of the following bacterial meningitis is associated with sudural effusion?? A. H. influenza. B. Neisseria meningitits. C. Streptococcus pneumonia. D. Enterococcus.

---

**Groundtruth**

**Explanation:**
Chronic Subdural Empyema It may be primary infection of subdural space from sinusitis focus causing suppuration and pus formation. It can be complication of the chronic subdural haematoma. It is due to secondary bacterial infection of collected clot/fluid. Infection is from sinusitis scalp (common)/through earlier trauma wound/haematogenous. Commonly Gram positive organisms cause empyema like streptococci (viridans/milleri) but other virulent organisms like Gram negative bacteria (H.influenzae) occasionally can cause. There is coical venous thrombophlebitis and coical infarction. Headache, fever, meningism and convulsions are the features. MRI is ideal than CT to diagnose. Treatment: Antibiotics, craniotomy and drainage; anticonvulsants, ICU care, proper monitoring, regular follow up. Condition has got 10% moality. Ref: SRB's Manual of Surgery 5th edition Pgno : 1097

**Answer: A**

---

**Output of strong model alone (Evaluator: 3.0)**

**Explanation:**
Subdural effusion, a complication of meningitis, is most commonly associated with Streptococcus pneumoniae. Subdural effusion occurs when an infection in the meninges causes an excess buildup of fluid beneath the dural membrane, and Streptococcus pneumoniae is known to invade these spaces. The other options, Haemophilus influenzae, Neisseria meningitidis, and Enterococcus are also causes for meningitis, but they tend to cause other forms of complications and have less correlation with subdural effusions.

**Answer: C**

---

**Positive Sample (Evaluator: 6.0)**

**Weak model output**

**Explanation:**
H. influenza type b is responsible for causing meningitis in children.\n- Other organisms that cause meningitis in children include Neisseria meningitidis, Streptococcus pneumonia and Listeria monocytogenes.\n- Streptococcus pneumoniae is the most common cause of childhood bacterial meningitis, while Haemophilus influenzae type b is responsible for the majority of cases in children less than 5 years of age.\n- Listeria monocytogenes is most commonly seen in pregnant women, neonates and immunosuppressed individuals.\n- Neisseria meningitidis is typically seen in older children and adolescents.

**Answer: A**

**Strong model output**

**Explanation:**
H. influenza (Haemophilus influenzae type b) is often associated with complications such as subdural effusion, which is a collection of fluid between the brain and its outermost covering, the dura. Other causative agents such as Neisseria meningitidis, Streptococcus pneumoniae, and Enterococcus are not typically associated with subdural effusion

**Answer: A**

---

**Negative Sample (Evaluator: 1.0)**

**Weak model output:**

**Explanation:**
Ans: c (Streptococcus pneumoniae)

**Answer: C**

**Strong model output**

**Explanation:**
Streptococcus pneumoniae (Answer C) is frequently associated with subdural effusion, a condition where fluid accumulates between the outermost and middle layers of the meninges - the protective layers covering the brain and spinal cord. Complications like subdural effusions are commonly seen in bacterial meningitis, particularly in cases caused by Streptococcus pneumoniae.

**Answer: C**

Figure 6: Case Study.

