# OpenReview forum: "Synergistic Weak-Strong Collaboration by Aligning Preferences"
_ICLR.cc/2025/Conference — ICLR 2025 Conference Withdrawn Submission_

### Official Review · Reviewer_uB8p · 2024-10-16

**Soundness:** 3
**Presentation:** 2
**Contribution:** 2
**Rating:** 5
**Confidence:** 4

**Summary:**

The authors propose a paradigm of weak-strong model cooperation, in which the large model with stronger reasoning ability is responsible for reasoning with the background knowledge and drafts generated by the small model. Furthermore, the authors propose to fine-tune the weak model to adapt it to the preferences of the strong model to achieve so-called adaptive cooperation. The proposed method achieves the improvement in the F1 performance in three datasets.

**Strengths:**

1. Considering the challenges of real-world scenarios, the issues that this paper focuses on are necessary. Strong-weak model collaboration is one of the promising directions.
2. "Using weak models for domain adaptation and then strong models for reasoning" can be seen as a RAG method in which the weak model after domain adaptation generates evidence context, and then the strong model uses the evidence in this domain for reasoning. This may actually increase the amount of information for strong model reasoning.

**Weaknesses:**

1. The authors explore the framework of weak-strong model cooperation, but I think it still needs to be better explained, that is, how the proposed feedback loop and interaction strategy go beyond the static cooperation method. I think the claims of L111-L115 are a bit far-fetched (considering that the weak model still reasons first during reasoning, and then the strong model uses the output of the weak model for reasoning). In addition, the writing needs to be improved, there are many small errors, and some claims are confusing to readers.
2. The paper focuses on improvements such as performance scores (F1), but lacks qualitative analysis of how the models collaborate in real-world scenarios. In fact, I am still confused about the example in Figure 6, how to show the role of the strong model? There is also limited information about how the feedback loop between weak and strong models affects the interpretability or usability of the output in complex reasoning tasks, but it is one of the important contributions emphasized by the authors. I suggest that the authors add some qualitative examples that can show how collaboration improves responses (in terms of factual accuracy, reasoning chain, or coherence).
3. The paper acknowledges the computational cost of fine-tuning large models, but the authors do not provide much insight into the scalability of COWEST when it is extended to larger weak models or more complex tasks, such as multi-hop questions that exploit the strong reasoning capabilities of large models. In addition, the resource impact of the feedback loop (e.g., computational overhead) is not discussed in depth, where the two inferences in the Inference stage increase the computational cost.
4. The authors should conduct comparative experiments on transferring domain knowledge to strong models in the case of longer contexts.

**Questions:**

1. The LLM abbreviation of L121 is repeatedly defined.
2. The reference form of L127-L128.
3. What if the weak model and the strong model are the same?

---

### Official Review · Reviewer_Ns8p · 2024-10-22

**Soundness:** 3
**Presentation:** 2
**Contribution:** 3
**Rating:** 5
**Confidence:** 4

**Summary:**

This work focuses on the challenge that current large language models (LLMs) often struggle with specific domains or downstream tasks. To tackle this, we propose a collaborative framework, CoWEST, which integrates a weak LLM with a strong LLM. In CoWEST, the weak LLM is first fine-tuned for a specific domain or task, and then the strong LLM’s general capabilities are leveraged to enhance the fine-tuned weak LLM’s output. Additionally, a preference tuning paradigm is used to evaluate the collaborative output against that of independent models. Extensive experiments demonstrate the effectiveness of the proposed CoWEST framework.

**Strengths:**

a) The proposed CoWEST shows remarkable improvements over SOTA methods such as RAG-based methods.

b) The interaction design between the weak LLM and the strong LLM is interesting compared to existing methods.

**Weaknesses:**

a) The sampling method for preference tuning is not clear, lack the the sampling statistics (e.g., sample distribution, average sample size etc.).

b) The evaluator is like a self-critique and more evaluator quality details such as score criteria, comparisons with human evaluation etc. should be included.

c) Minor writing issues.

**Questions:**

a) Do the authors have a vision on how the proposed CoWEST is different from the LLM cascade methods such as CAT[1]?

b) How the sampling will impact on the performance?

c) How's the evaluator's quality? Have the author consider using logits or a trainable method (e.g., a MLP) to serve as the evaluator? Since self-critique sometimes may results LLM is always more confident with the content generated by itself, while logits or trainable methods can be more fair.

d) In 127-129, using \citep()

e) In line 207, "referred to as"? there are more, please check the writing for readibility.

**Reference**

[1] Cascade-Aware Training of Language Models.

---

### Official Review · Reviewer_wAwg · 2024-10-27

**Soundness:** 1
**Presentation:** 2
**Contribution:** 1
**Rating:** 3
**Confidence:** 4

**Summary:**

The paper proposed a weak-strong collaboration mode, in which a weak model fine-tuned on domain-specific datasets first generates drafts, while a strong model refines them. By utilizing feedback from the strong model to perform preference optimization, the performance of the weak model is further improved.

**Strengths:**

The research topic regarding the collaborative interaction between a specialized weak model and a general strong model is very important

**Weaknesses:**

1. Lack of novelty: The concept of weak-strong collaboration explored in the paper, essentially using feedback to correct large language models, is not a novel idea and has already been extensively researched [1]. The two collaboration strategies: standard refinement bears strong resemblances to prior works [2], and preference enhancement that leverages DPO for inconsistency alignment is also not new. It’s just old wine in a new bottle, wrapping up a story of the interaction between a specialized weak model and a general strong model.
2. The datasets used in the experiments lack representativeness: (1) Domain selection: In addition to the three domains selected, more typical mathematical reasoning datasets should be included, such as GSM8k and MATH, which have been widely used in previous model collaboration work [3][4]. (2) Dataset selection: For the medical domain, the choice of MedMCQA, which is limited to a multiple-choice format, is too narrow. There should be more focus on broader and more practical long-form QA datasets like K-QA [5].
3. Lack of baselines for model collaboration/ensemble: The main experiment mainly compares the proposed collaboration approach with only weak or strong model strategies, omitting critical baseline comparisons, such as self-refine [6], and other ensemble strategies such as multi-model debate [7], self-consistency.
4. Some specific experimental settings were not clearly stated, for example, the retrieval knowledge base used by FLARE in three selected domains was not mentioned
5. The Preference Enhancement Interaction lacks generalizability, as the acquisition of preference pairs is specific to a strong model. This specificity might limit the effectiveness and generalization when collaborating with different strong models.
6. Questioning the experimental results: The results presented in Table 1 raise concerns about the necessity of weak-strong collaboration. In the Counterfactual and Medicine domains, weak models without SFT are much stronger than strong models, e.g., Llama-3-8b (68.57) vs. GPT-3.5-turbo (22.62). Similarly, in the Ethics domain, the performances were comparable. If weak models can perform on par with or better than strong models, is the use of weak-strong collaboration justified? Does the motivation for using a stronger model to assist weaker ones still stand?
7. Concerns about the high costs for strong models compared to minor performance improvements in weak models: The proposed collaborative approach, compared to merely using a weak model for SFT, only brought minor improvements (shown in Table 1). However, this process requires the strong model to refine and evaluate the output of the weak model, which brings significant API costs.
8. Lack of in-depth analysis of the improvements brought by the cooperation strategy, for example, the paper does not specify in which aspects the strong model has improved the weak model, nor does it detail the types and percentages of errors detected in the weak model by the strong model. Furthermore, the frequency with which the weak model adopts feedback from the strong model is not discussed. More comprehensive case studies are needed to understand these dynamics fully, rather than merely providing a superficial overview.

[1] Automatically Correcting Large Language Models: Surveying the Landscape of Diverse Automated Correction Strategies. Pan et al. TACL 2024

[2] Small Models are Valuable Plug-ins for Large Language Models. Xu et al. ACL 2024 Findings

[3] Learning to Decode Collaboratively with Multiple Language Models. Shen et al. ACL 2024

[4] Ensemble learning for heterogeneous large language models with deep parallel collaboration. Huang et al. NeurIPS 2024

[5] K-QA: A Real-World Medical Q&A Benchmark. Manes et al. BioNLP 2024

[6] Self-Refine: Iterative Refinement with Self-Feedback. Madaan et al. NeurIPS 2023

[7] Improving Factuality and Reasoning in Language Models through Multiagent Debate. Du et al. arXiv 2023

**Questions:**

1. Why does the main experiment use the strong model GPT-3.5-Turbo for the ethical dataset, instead of maintaining consistency with other domains by using GPT-4?
2. Why was the learning rate set to 1.41e-5? Intuitively, this seems like an uncommon number, was it determined by searching different learning rates?
3. Typo: There is inconsistent formatting of the name 'Llama-3' throughout the paper. For example, it is written as "LLama-3-8B" in Table 1, "LLaMA3-8B" on line 481, and "Llama3-8B" on line 381.
4. In the main experiment, were the results for Llama-3-8B obtained using a few-shot setting? The IfQA paper used two evaluation methods: a supervised setting and a few-shot setting. If the few-shot setting was not used, intuitively, the output form of the model might not be controllable. Similarly, when using Llama-3-70B and Llama-2-70B as strong models for evaluation, were few-shot settings adopted?

---

### Official Review · Reviewer_pkSH · 2024-10-30

**Soundness:** 2
**Presentation:** 3
**Contribution:** 2
**Rating:** 3
**Confidence:** 4

**Summary:**

This paper presents a collaborative framework that integrates a specialized weak model with a general strong model to enhance the reasoning performance of LLMs. In this framework, the weak model generates detailed initial drafts and background information tailored to specific domains, while the strong model refines and enhances these drafts utilizing its advanced reasoning capabilities. A feedback loop is implemented to fine-tune the weak model based on the preferences of the strong model, fostering an adaptive and synergistic relationship. Experimental results indicate that the proposed method outperforms both the basic weak and strong LLMs.

**Strengths:**

1. This paper is well-organized and easy to read.
2. The proposed method presents a reasonable approach to improve the reasoning performance of LLMs by combining weak and strong LLMs.
3. The approach is practical and has the potential for broad application.
4. The experimental results reveal that the proposed method significantly enhances performance on various reasoning tasks compared to both the weak and strong LLMs.

**Weaknesses:**

1. The technical innovations introduced in this paper appear to be somewhat limited, as the concept of leveraging both weak and strong LLMs has been extensively explored in prior research, including works such as “Your Weak LLM is Secretly a Strong Teacher for Alignment” and “Synthesizing Text-to-SQL Data from Weak and Strong LLMs.”
2. A more comprehensive evaluation would enhance the study by comparing the proposed method against a more comprehensive array of advanced baseline models. Currently, the comparisons are limited to several basic baselines. Incorporating more sophisticated weak-strong collaboration methods and state-of-the-art techniques would provide stronger validation of the proposed method's effectiveness.
3. To demonstrate the versatility of the proposed method, it would be advantageous to conduct experiments using different open-source LLMs of varying sizes.

**Questions:**

Please refer to the Weaknesses.

---

### Note · Authors · 2024-11-25

I have read and agree with the venue's withdrawal policy on behalf of myself and my co-authors.